# Rethinking the Construction of Effective Metrics for Understanding the Mechanisms of Pretrained Language Models

**You Li**[*]    **Jinhui Yin**[*]    **Yuming Lin**[†]

Guangxi Key Laboratory of Trusted Software, Guilin University of Electronic Technology

liyou@guet.edu.cn    yinjinhui55@gmail.com    ymlin@guet.edu.cn

## Abstract

Pretrained language models are expected to effectively map input text to a set of vectors while preserving the inherent relationships within the text. Consequently, designing a white-box model to compute metrics that reflect the presence of specific internal relations in these vectors has become a common approach for post-hoc interpretability analysis of pretrained language models. However, achieving interpretability in white-box models and ensuring the rigor of metric computation becomes challenging when the source model lacks inherent interpretability. Therefore, in this paper, we discuss striking a balance in this trade-off and propose a novel line to constructing metrics for understanding the mechanisms of pretrained language models. We have specifically designed a family of metrics along this line of investigation, and the model used to compute these metrics is referred to as the tree topological probe. We conducted measurements on BERT-large by using these metrics. Based on the experimental results, we propose a speculation regarding the working mechanism of BERT-like pretrained language models, as well as a strategy for enhancing fine-tuning performance by leveraging the topological probe to improve specific submodules.[1]

## 1 Introduction

Pretrained language models consisting of stacked transformer blocks (Vaswani et al., 2017) are commonly expected to map input text to a set of vectors, such that any relationship in the text corresponds to some algebraic operation on these vectors. However, it is generally unknown whether such operations exist. Therefore, designing a white-box model that computes a metric for a given set of vectors corresponding to a text, which reflects to

some extent the existence of operations extracting specific information from the vectors, is a common approach for post-hoc interpretability analysis of such models (Maudslay et al., 2020; Limisiewicz and Marecek, 2021; Chen et al., 2021; White et al., 2021; Immer et al., 2022). However, even though we may desire strong interpretability from a white-box model and metrics computed by it that rigorously reflect the ability to extract specific information from a given set of vectors, it can be challenging to achieve both of these aspects simultaneously when the source model lacks inherent interpretability. Therefore, making implicit assumptions during metric computation is common (Kornblith et al., 2019; Wang et al., 2022). A simple example is the cosine similarity of contextual embeddings. This metric is straightforward and has an intuitive geometric interpretation, making it easy to explain, but it tends to underestimate the similarity of high-frequency words (Zhou et al., 2022).

On the other hand, due to the intuition that 'if a white-box model cannot distinguish embeddings that exhibit practical differences (such as context embeddings and static embeddings), it should be considered ineffective,' experimental validation of a white-box model's ability to effectively distinguish between embeddings with evident practical distinctions is a common practice in research. Furthermore, if the magnitude of metrics computed by a white-box model strongly correlates with the quality of different embeddings in practical settings, researchers usually trust its effectiveness. Therefore, in practice, traditional white-box models actually classify sets of vectors from different sources.

Taking the structural probe proposed by Hewitt and Manning as an example, they perform a linear transformation on the embedding of each complete word in the text and use the square of the L2 norm of the transformed vector as a prediction for the depth of the corresponding word in the dependency tree (Hewitt and Manning, 2019). In this way, the

---

*Equal contribution.

†Corresponding Author.

[1]Our code is available at https://github.com/cclx/Effective_Metrics

linear transformation matrix serves as a learning parameter, and the minimum risk loss between the predicted and true depths is used as a metric. Intuitively, the smaller the metric is, the more likely the embedding contains complete syntax relations. The experimental results indeed align with this intuition, showing that contextual embeddings (such as those generated by BERT (Devlin et al., 2019)) outperform static embeddings. However, due to the unknown nature of the true deep distribution, it is challenging to deduce which geometric features within the representations influence the magnitude of structural probe measurements from the setup of structural probe. Overall, while the results of the structural probe provide an intuition that contextual embeddings, such as those generated by BERT, capture richer syntactic relations than those of the traditional embeddings, it is currently impossible to know what the geometric structure of a "good" embedding is for the metric defined by the structural probe.

In addition, to enhance the interpretability and flexibility of white-box models, it is common to include assumptions that are challenging to empirically validate. For example, Ethayarajh proposed to use anisotropy-adjusted self-similarity to measure the context-specificity of embeddings (Ethayarajh, 2019). Since the computation of this metric doesn't require the introduction of additional human labels, it is theoretically possible to conduct further analysis, such as examining how fundamental geometric features in the representation (e.g., rank) affect anisotropy-adjusted self-similarity, or simply consider this metric as defining a new geometric feature. Overall, this is a metric that can be discussed purely at the mathematical level. However, verifying whether the measured context-specificity in this metric aligns well with context-specificity in linguistics, without the use of, or with only limited additional human labels, may be challenging. Additionally, confirming whether the model leverages the properties of anisotropy-adjusted self-similarity during actual inference tasks might also be challenging.

There appears to be a trade-off here between two types of metrics:

1. Metrics that are constrained by supervised signals with ground truth labels, which provide more practical intuition.

2. Metrics that reflect the geometric properties of the vector set itself, which provide a more formal representation.

Therefore, we propose a new line that takes traditional supervised probes as the structure of the white-box model and then self-supervises it, trying to preserve both of the abovementioned properties as much as possible. The motivation behind this idea is that any feature that is beneficial for interpretability has internal constraints. If a certain feature has no internal constraints, it must be represented by a vector set without geometric constraints, which does not contain any interpretable factors. Therefore, what is important for interpretability is the correspondence between the internal constraints of the probed features and the vector set, which can describe the geometric structure of the vector set to some extent. **In the case where the internal constraints of the probed features are well defined, a probe that detects these features can naturally induce a probe that detects the internal constraints, which is self-supervised**.

In summary, the contributions of this work include:

1. We propose a novel self-supervised probe, referred to as the **tree topological probe**, to probe the hierarchical structure of sentence representations learned by pretrained language models like BERT.

2. We discuss the theoretical relationship between the tree topological probe and the structural probe, with the former bounding the latter.

3. We measure the metrics constructed based on the tree topological probe on BERT-large. According to the experimental results, we propose a speculation regarding the working mechanism of a BERT-like pretrained language model.

4. We utilize metrics constructed by the tree topological probe to enhance BERT's submodules during fine-tuning and observe that enhancing certain modules can improve the fine-tuning performance. We also propose a strategy for selecting submodules.

## 2   Related Work

The probe is the most common approach for associating neural network representations with linguistic properties (Voita and Titov, 2020). This approach is widely used to explore part of speech

knowledge (Belinkov and Glass, 2019; Voita and Titov, 2020; Pimentel et al., 2020; Hewitt et al., 2021) and for sentence and dependency structures (Hewitt and Manning, 2019; Maudslay et al., 2020; White et al., 2021; Limisiewicz and Marecek, 2021; Chen et al., 2021). These studies demonstrate many important aspects of the linguistic information are encoded in pretrained representations. However, in some probe experiments, researchers have found that the probe precision obtained by both random representation and pretrained representation were quite close (Zhang and Bowman, 2018; Hewitt and Liang, 2019). This demonstrates that it is not sufficient to use the probe precision to measure whether the representations contain specific language information. To improve the reliability of probes, some researchers have proposed the use of control tasks in probe experiments (Hewitt and Liang, 2019). In recent research, Lovering et al. realized that inductive bias can be used to describe the ease of extracting relevant features from representations. Immer et al. further proposed a Bayesian framework for quantifying inductive bias with probes, and they used the Model Evidence Maximum instead of trivial precision.

## 3    Methodology

As the foundation of the white-box model proposed in this paper is built upon the traditional probe, we will begin by providing a general description of the probe based on the definition presented in (Ivanova et al., 2021). Additionally, we will introduce some relevant notation for better understanding.

### 3.1    General Form of the Probe

Given a character set, in a formal language, the generation rules uniquely determine the properties of the language. We assume that there also exists a set of generation rules $\mathcal{R}$ implicitly in natural language, and the language objects derived from these rules exhibit a series of features. Among these features, a subset $Y$ is selected as the probed feature for which the properties represent the logical constraints of the generation rule set. Assuming there is another model $\mathcal{M}$ that can assign a suitable representation vector to the generated language objects, the properties of $Y$ are then represented by the intrinsic geometric constraints of the vector set. By studying the geometric constraints that are implicit in the vector set and that correspond to $Y$, especially when $Y$ is expanded to all features of the language object,

we can determine the correspondence between $\mathcal{M}$ and $\mathcal{R}$. The probe is a model that investigates the relationship between the geometric constraints of the vector set and $Y$. It is composed of a function set $F$ and a metric $E_Y$ defined on $Y$. The input of a function in $F$ is the representation vector of a language object, and the output is the predicted $Y$ feature of the input language object. The distance between the predicted feature and the true feature is calculated by using the metric $E_Y$, and a function $f$ in $F$ that minimizes the distance is determined. Here, $F$ limits the range of geometric constraints, and $E_Y$ limits the selection of a "good" geometry. Notably, this definition seems very similar to that of learning. Therefore, the larger the scope of $F$ is, the harder it is to discern the form of the geometric constraints, especially when $F$ is a neural network (Pimentel et al., 2020; White et al., 2021). However, the purpose of the probe is different from that of learning. The goal of learning is to construct a model $\mathcal{M}$ (usually a black box), which may have multiple construction methods, while the purpose of the probe is to analyze the relationship between $\mathcal{M}$ and $\mathcal{R}$.

### 3.2    The Design Scheme for the Topological Probe

One of the goals of topology is to find homeomorphic or homotopic invariants (including invariant quantities, algebraic structures, functors, etc.) and then to characterize the intrinsic structure of a topological space with these invariants. Analogously, we can view $R$ as a geometric object and $Y$ as its topology. Can we then define a concept similar to topological invariants with respect to $Y$?

We define a feature invariant for $Y$ as a set of conditions $C_Y$ such that any element in $Y$ satisfies $C_Y$. $C_Y$ reflects the internal constraints of the probed feature, as well as a part of the logical constraints of $R$. Furthermore, if $C_Y$ is well defined, it induces a set $X_{C_Y}$ consisting of all objects satisfying $C_Y$, which naturally extends the metric defined on $Y$ to $X_{C_Y}$.

Furthermore, just as the distance measure between two points can induce a distance measure between a point and a plane, the distance measure between the predicted feature $px$ and $X_{C_Y}$ can also be induced by $E_Y$ (denoted as $E_{C_Y}$):

$$E_{C_Y}(px, X_{C_Y}) = \min_{x \in X_{C_Y}} E_Y(px, x) \quad (1)$$

It can be easily verified that if $E_Y$ is a

well-defined distance metric on $Y$, then $E_{C_Y}$ should also be a well-defined distance metric on $px$. Once we have $E_{C_Y}$, the supervised probe $(F, E_Y, Y)$ can naturally induce a self-supervised probe $(F, E_{C_Y}, C_Y)$. We refer to $(F, E_{C_Y}, C_Y)$ as the self-supervised version of $(F, E_Y, Y)$, also known as the topological probe.

Notably, the prerequisite for obtaining $(F, E_{C_Y}, C_Y)$ is that $C_Y$ must be well-defined, so $C_Y$ should not be a black box. Figure 1 shows an intuitive illustration.

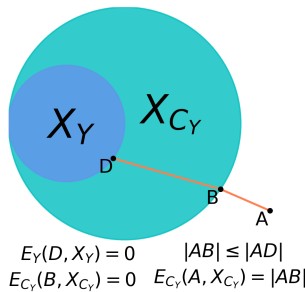

$$E_Y(D, X_Y) = 0 \qquad |AB| \le |AD|$$
$$E_{C_Y}(B, X_{C_Y}) = 0 \qquad E_{C_Y}(A, X_{C_Y}) = |AB|$$

Figure 1: The relationship between the distance from the predicted feature A to $X_{C_Y}$ and the distance from A to $X_Y$.

Next, we present a specific topological probe that is based on the previously outlined design scheme and serves as a self-supervised variant of the structural probe.

### 3.3 The Self-supervised Tree Topological Probe

Given a sentence $W$, it is represented by a model $M$ as a set (or sequence) of vectors, denoted as $H = M(W)$. The number of vectors in $H$ is denoted as $L_H$, and we assign an index $(1, 2, ...L_H)$ to each vector in $H$ so that the order of the indices matches the order of the corresponding tokens in the sentence. Additionally, we denote the dimension of the vectors as $n$. For each $W$, there exists a syntax tree $T_W$, where each complete word in $W$ corresponds to a node in $T_W$.

The probed feature $Y$ that the structural probe defines is the depth of the nodes corresponding to complete words. Following the work in (Hewitt and Manning, 2019), we set the parameter space of $F$ for the structural probe to be all real matrices of size $m * n$, where $m < n$. The specific form for predicting the depth is as follows: Given

$p \in R, \forall 1 \le i \le L_H$

$$\boldsymbol{p}dep(h_i) = \|f * h_i\|^p \qquad (2)$$

where $\boldsymbol{p}dep(h_i)$ is the prediction tree depth of $w_i$ in $T_W$ and $f$ is a real matrix of size $m * n$. Because $\forall p < 2$, there is a tree that cannot be embedded as above (Reif et al., 2019), so $p$ is usually taken as 2. $\boldsymbol{p}dep(h_1), \boldsymbol{p}dep(h_2) \cdots, \boldsymbol{p}dep(h_{L_H})$ form a sequence denoted as $\boldsymbol{p}dep_H$.

Moreover, we denote the true depth of $w_i$ as $dep(w_i)$. Hence, $dep(w_1), dep(w_2) \cdots, dep(w_{L_H})$ also form a sequence denoted as $dep_W$. The metric $E$ in the structural probe is defined as follows:

$$E(\boldsymbol{p}dep_H, dep_W)$$

$$= \frac{1}{L_H} \sum_{i=1}^{L_H} (\boldsymbol{p}dep(h_i) - dep(w_i))^2 \qquad (3)$$

Therefore, the structural probe is defined as $(|f * |^2, E, dep)$.

Now we provide the constraints $C_{dep}$ for $dep$. An important limitation of $dep_W$ is that it is an integer sequence. Based on the characteristics of the tree structure, it is naturally determined that $dep_W$ must satisfy the following two conditions:

**(Boundary condition).** If $L_H \ge 1$, there is exactly one minimum element in $dep_W$, and it is equal to 1; if $L_H \ge 2$, at least one element in $dep_W$ is equal to 2.

**(Recursion condition).** If we sort $dep_W$ in ascending order to obtain the sequence $asdep_W$, then

$$\forall 1 \le i \le L_H - 1$$

$$asdep(w_{i+1}) = asdep(w_i)$$

or

$$asdep(w_{i+1}) = asdep(w_i) + 1$$

We denote the set of all sequences that conform to $C_{dep}$ as $X_{C_{dep}}$. From equation 1, we can induce a metric $E_{C_{dep}}$:

$$E_{C_{dep}}(\boldsymbol{p}dep_H, X_{C_{dep}}) = \min_{x \in X_{C_{dep}}} E(\boldsymbol{p}dep_H, x)$$
$$(4)$$

Assuming we can construct an explicit sequence $mins_W$ such that:

$$mins_W = \arg\min_{x \in X_{C_{dep}}} \sum_{i=1}^{L_H} (\boldsymbol{p}dep(h_i) - x(w_i))^2 \quad (5)$$

We can obtain an analytical expression for $E_{C_{dep}}$ as follows:

$$E_{C_{dep}}(\boldsymbol{p}dep_H, X_{C_{dep}}) = E(\boldsymbol{p}dep_H, mins_W) \tag{6}$$

Consider the following two examples:

1. When $\boldsymbol{p}dep_H = 0.8, 1.5, 1.8, 2.4, 4.5$, then $mins_W = 1, 2, 2, 3, 4$.

2. When $\boldsymbol{p}dep_H = 0.8, 1.5, 1.8, 2.4, 7.5$, then $mins_W = 1, 2, 3, 4, 5$.

It can be observed that the predicted depths for nodes further down the hierarchy can also influence the corresponding values of $mins_W$ for nodes higher up in the hierarchy. In the examples provided, due to the change from 4.5 to 7.5, 1.8 changes from 2 to 3 at the corresponding $mins_W$. Therefore, using a straightforward local greedy approach may not yield an accurate calculation of $mins_W$, and if a simple enumeration method is employed, the computational complexity will become exponential.

However, while a local greedy approach may not always provide an exact computation of $mins_W$, it can still maintain a certain degree of accuracy for reasonable results of $\boldsymbol{p}dep_H$. This is because cases like the jump from 2.4 to 7.5 should be infrequent in a well-trained probe's computed sequence of predicted depths, unless the probed representation does not encode the tree structure well and exhibits a disruption in the middle.

Before delving into that, we first introduce some notations:

- $\boldsymbol{ap}dep_H$ denote the sequence obtained by sorting $\boldsymbol{p}dep_H$ in ascending order.

- $\boldsymbol{ap}dep_i$ represents the $i$-th element of $\boldsymbol{ap}dep_H$.

- $pre_W$ be a sequence in $X_{C_{dep}}$.

Here, we introduce a simple method for constructing $mins_W$ from a local greedy perspective.

**(Initialization).** If $L_H \geq 1$, let $pre(w_1) = 1$; if $L_H \geq 2$, let $pre(w_2) = 2$.

**(Recurrence).** If $L_H \geq 3$ and $3 \leq i \leq L_H$, let

$$pre(w_i) = pre(w_{i-1}) + bias_{i-1} \tag{7}$$

where the values of $bias_{i-1}$ and $\boldsymbol{ap}dep_H$ are related if

$$|pre(w_{i-1}) + 1 - \boldsymbol{ap}dep_i| \leq |pre(w_{i-1}) - \boldsymbol{ap}dep_i|$$

$bias_{i-1} = 1$; otherwise, $bias_{i-1} = 0$.

**(Alignment).** Let $a_i(1 \leq i \leq L_H)$ denote the index of $\boldsymbol{ap}dep_i$ in $\boldsymbol{p}dep_H$. Then, let

$$pesu(w_{a_i}) = pre(w_i) \tag{8}$$

It can be shown that $pesu_W$ constructed in the above manner satisfies the following theorem:

**Theorem 1.** *If* $\forall i = 1, 2 \cdots, L_H - 1$, $\boldsymbol{ap}dep_{i+1} - \boldsymbol{ap}dep_i <= 1$, *then*

$$E(\boldsymbol{p}dep_H, pesu_W) = E(\boldsymbol{p}dep_H, mins_W)$$

Therefore, $pesu_W$ can be considered an approximation to $mins_W$. Appendix A contains the proof of this theorem. In the subsequent sections of this paper, we replace $E_{C_{dep}}(\boldsymbol{p}dep_H, X_{C_{dep}})$ with $E(\boldsymbol{p}dep_H, pesu_W)$.

Additionally, an important consideration is determining the appropriate value of the minimum element for $dep_W$ in the boundary condition. In the preceding contents, we assumed a root depth of 1 for the syntactic tree. However, in traditional structural probe (Hewitt and Manning, 2019; Maudslay et al., 2020; Limisiewicz and Marecek, 2021; Chen et al., 2021; White et al., 2021), the root depth is typically assigned as 0 due to the annotation conventions of syntactic tree datasets. From a logical perspective, these two choices may appear indistinguishable.

However, in Appendix B, we demonstrate that the choice of whether the root depth is 0 has a significant impact on the geometry defined by the tree topological probe. Furthermore, we can prove that as long as the assigned root depth is greater than 0, the optimal geometry defined by the tree topological probe remains the same to a certain extent. Therefore, in the subsequent sections of this paper, we adopt the setting where the value of the minimum element of $dep_W$ is 1.

### 3.4 Enhancements to the Tree Topological Probe

Let the set of all language objects generated by rule $R$ be denoted as $\mathcal{X}_R$, and the cardinality of $\mathcal{X}_R$ be denoted as $|\mathcal{X}_R|$. The structural probe induces a metric that describes the relationship between model $M$ and $dep$:

$$\mathcal{X}_{sp}(M) = \min_{f \in F} \frac{1}{|\mathcal{X}_R|} \sum_{W \in \mathcal{X}_R} E(\boldsymbol{p}dep_{M(W)}, dep_W) \tag{9}$$

The tree topological probe can also induce a similar metric:

$$\mathcal{X}_{ssp}(M) =$$

$$\min_{f \in F} \frac{1}{|\mathcal{X}_R|} \sum_{W \in \mathcal{X}_R} E(\boldsymbol{p}dep_{M(W)}, mins_W) \quad (10)$$

On the other hand, we let

$$maxs_W = \arg\max_{x \in X_{C_{dep}}} \sum_{i=1}^{L_H} (\boldsymbol{p}dep(h_i) - x(w_i))^2 \quad (11)$$

similar to $mins_W$, and $maxs_W$, inducing the following metrics:

$$\mathcal{X}_{essp}(M)$$

$$= \min_{f \in F} \frac{1}{|\mathcal{X}_R|} \sum_{W \in \mathcal{X}_R} E(\boldsymbol{p}dep_{M(W)}, maxs_W) \quad (12)$$

Since $dep_W \in X_{C_{dep}}$, when $f$ is given, we have:

$$E(\boldsymbol{p}dep_{M(W)}, dep_W) \leq \max_{x \in X_{C_{dep}}} E(\boldsymbol{p}dep_{M(W)}, x)$$

Furthermore, as $\mathcal{X}sp(M)$ and $\mathcal{X}essp(M)$ share the same set of probing functions $F$, we have:

$$\mathcal{X}_{sp}(M) \leq \mathcal{X}_{essp}(M)$$

Therefore, $\mathcal{X}_{essp}(M)$ provides us with an upper bound for the structural probe metric. Similarly, for $\mathcal{X}ssp(M)$, we also have:

$$\mathcal{X}_{ssp}(M) \leq \mathcal{X}_{sp}(M)$$

Therefore, $\mathcal{X}_{ssp}(M)$ provides us with a lower bound for the structural probe metric. In summary, we have the following:

$$\mathcal{X}_{ssp}(M) \leq \mathcal{X}_{sp}(M) \leq \mathcal{X}_{essp}(M)$$

If $\mathcal{X}ssp(M) = \mathcal{X}essp(M)$, then there is no difference between the tree topological probe and the structural probe. On the other hand, if it is believed that a smaller $\mathcal{X}sp(M)$ is desirable, then estimating $\mathcal{X}sp(M)$ within the range $[\mathcal{X}ssp(M), \mathcal{X}essp(M)]$ becomes an interesting problem. We consider the following:

$$\theta_W =$$

$$\frac{E(\boldsymbol{p}dep_{M(W)}, dep_W) - E(\boldsymbol{p}dep_{M(W)}, mins_W)}{E(\boldsymbol{p}dep_{M(W)}, maxs_W) - E(\boldsymbol{p}dep_{M(W)}, mins_W)} \quad (13)$$

This leads to an intriguing linguistic distribution, the distribution of $\theta_W \in [0, 1]$ when uniformly

sampling $W$ from $\mathcal{X}_R$. We suppose the density function of this distribution is denoted as $P_\theta$, and the expectation with respect to $\theta$ is denoted as $E_{P_\theta}$. Then we can approximate $\mathcal{X}sp(M)$ as follows:

$$\mathcal{X}sp(M) = E_{P_\theta} \mathcal{X}essp(M) + (1 - E_{P_\theta}) \mathcal{X}ssp(M) \quad (14)$$

While the analysis of $P_\theta$ is not the primary focus of this paper, in the absence of any other constraints or biases on model $M$, we conjecture that the distribution curve of $\theta$ may resemble a uniform bell curve. Hence, we consider the following distribution approximation:

$$P_\theta(x) = 6(x - x^2) \quad x \in [0, 1]$$

At this point:

$$\mathcal{X}sp(M) = \frac{1}{2}(\mathcal{X}essp(M) + \mathcal{X}ssp(M)) \quad (15)$$

Therefore, utilizing a self-supervised metric can approximate the unbiased optimal geometry defined by the structural probe:

$$M_G = \arg\min_M \frac{1}{2}(\mathcal{X}essp(M) + \mathcal{X}ssp(M)) \quad (16)$$

Moreover, $M_G$ is an analytically tractable object, implying that the metrics induced by the tree topological probe preserve to a certain extent the two metric properties discussed in the introduction. However, there is a crucial issue that remains unresolved. Can we explicitly construct $maxs_W$? Currently, we have not found a straightforward method similar to constructing $pesu_W$ for approximating $maxs_W$. However, based on the sorting inequality, we can construct a sequence that approximates $maxs_W$ based on $pre_W$. Let $d_i (1 \leq i \leq L_H)$ denote $L_H - i + 1$. Then, let

$$xpesu(w_{a_i}) = pre(w_{d_i}) \quad (17)$$

In our subsequent experiments, we approximate $E(\boldsymbol{p}dep_H, maxs_W)$ with $E(\boldsymbol{p}dep_H, xpesu_W)$.

## 4 Experiments

In this section, delve into a range of experiments conducted on the tree topological probe, along with the underlying motivations behind them. To accommodate space limitations, we include many specific details of the experimental settings in Appendices C and D. Moreover, we focus our experiments on BERT-large and its submodules. Moreover, conducting similar experiments on other models is also straightforward (refer to Appendix F for supplementary results of experiments conducted using RoBERTa-large).

## 4.1 Measuring $\mathcal{X}ssp$ and $\mathcal{X}essp$ on BERT

We denote the model consisting of the input layer and the first $i$ transformer blocks of BERT-large as $M_i (0 \leq i \leq 24)$. Since the input of $M_i$ consists of tokenized units, including special tokens [CLS], [SEP], [PAD], and [MASK], we can conduct at least four types of measurement experiments:

e1. Measurement of the vector set formed by token embedding and special token embedding.

e2. Measurement of the vector set formed solely by token embedding.

e3. Measurement of the vector set formed by estimated embedding of complete words using token embedding and special token embedding.

e4. Measurement of the vector set formed solely by estimated embedding of complete words using token embedding.

Similarly, due to space constraints, we focus on discussing e1 in this paper. The measurement results are shown in Tables 1 and 2. The precise measurement values can be found in Appendix E. Furthermore, as shown in Figure 2, we present the negative logarithm curves of three measurement values as a function of $M_i$ variation.

| $\mathcal{X}ssp$ | $M$ |
|---|---|
| 0.01~0.05 | $M_0 \sim M_{11}$ |
| 0.05~0.1 | $M_{12} \sim M_{21}$ |
| 0.1~0.15 | $M_{22} \sim M_{23}$ |

Table 1: Grouping $M_i$ based on $\mathcal{X}ssp$. $M_l \sim M_r$ denotes $M_l, M_{l+1}, M_{l+2}, \ldots, M_r$. For example, the first row of the table indicates that the exact values of $\mathcal{X}ssp$ for $M_0, M_1, M_2, \ldots, M_{11}$ fall within the range of 0.01 to 0.05.

| $\mathcal{X}essp$ | $M$ | | |
|---|---|---|---|
| 0.3~0.4 | $M_3 \sim M_4$ | $M_7 \sim M_{12}$ | |
| 0.4~0.5 | | $M_{13} \sim M_{14}$ | |
| 0.5~1.0 | $M_1 \sim M_2$ | $M_5 \sim M_6$ | $M_{15} \sim M_{19}$ |
| 1.0~2.0 | | | $M_{20} \sim M_{24}$ |
| $\geq 4.0$ | $M_0$ | | |

Table 2: Grouping $M_i$ based on $\mathcal{X}essp$. Similar to the explanation in the caption of Table 1.

By examining the experimental results presented above, we can ascertain the following findings:

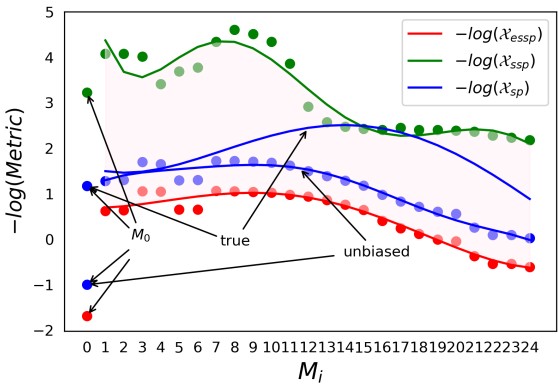

Figure 2: Negative logarithm of $\mathcal{X}_{ssp}$, $\mathcal{X}_{essp}$, unbiased $\mathcal{X}_{sp}$ and true $\mathcal{X}_{sp}$ across $M_i$.

f1. $\mathcal{X}ssp$ and $\mathcal{X}essp$ indeed bound the actual $\mathcal{X}sp$, and for $M_{14}$ to $M_{18}$, their true $\mathcal{X}sp$ are very close to their $\mathcal{X}ssp$.

f2. $M_0$ serves as a good baseline model. Furthermore, using $\mathcal{X}essp$ and unbiased $\mathcal{X}sp$ allows for effective differentiation between embeddings generated by models consisting solely of the regular input layer and those generated by models incorporating transformer blocks.

f3. For $M_1$ to $M_6$, their true $\mathcal{X}sp$ are very close to their unbiased $\mathcal{X}sp$.

f4. Both the curve of $-log(\mathcal{X}essp)$ and the curve of the true $-log(\mathcal{X}sp)$ follow an ascending-then-descending pattern. However, the models corresponding to their highest points are different, namely, $M_8$ and $M_{16}$, respectively.

f5. For the curve of $-log(\mathcal{X}ssp)$, its overall trend also shows an ascending-then-descending pattern but with some fluctuations in the range of $M_3$ to $M_6$. However, the model corresponding to its highest point is consistent with $-log(\mathcal{X}essp)$, which is $M_8$.

f6. The true $\mathcal{X}sp$ does not effectively distinguish between $M_0$ and $M_1$.

Based on the above findings, we can confidently draw the following rigorous conclusions:

c1. Based on f1, we can almost infer that $dep_W \in \underset{x \in X_{C_{dep}}}{\arg\min} \sum_{i=1}^{L_H} (\boldsymbol{p}dep(h_i) - x(w_i))^2$ for $M_{14}$ to $M_{18}$. **This implies that they memorize the preferences of the real data and minimize as much as possible to approach the**

**theoretical boundary**. Building upon f5, we can further conclude that the cost of memorizing $dep_W$ is an increase in $\mathcal{X}ssp$, which leads to a decrease in the accuracy of the embedding's linear encoding for tree structures.

c2. Based on f1, we can conclude that there exists a model $M$ where the true $\mathcal{X}sp(M)$ aligns with the $\mathcal{X}ssp(M)$ determined by $C_{dep}$. **This indicates that $C_{dep}$ serves as a sufficiently tight condition**.

c3. Based on f3, we can infer that $M_1$ to $M_6$ may not capture the distributional information of the actual syntactic trees, resulting in their generated embeddings considering only the most general case for linear encoding of tree structures. This implies that the distribution curve of their $\theta_W$ parameters is uniformly bell-shaped.

c4. Based on f2 and f6, we can conclude that the tree topological probe provides a more fine-grained evaluation of the ability to linearly encode tree structures in embedding vectors compared to the structural probe.

c5. Based on f3, f4 and f5, we can conclude that in BERT-large, embedding generated by $M_8$ and its neighboring models exhibit the strongest ability to linearly encode tree structures. Moreover, they gradually start to consider the distribution of real dependency trees, resulting in the true $\mathcal{X}sp(M)$ approaching $\mathcal{X}ssp(M)$ until reaching $M_{16}$.

c6. Based on f4 and f5, we can conclude that starting from $M_{16}$, the embeddings generated by $M_i$ gradually lose their ability to linearly encode tree structures. The values of $\mathcal{X}ssp$ and $\mathcal{X}essp$ for these models are generally larger compared to models before $M_{16}$. However, they still retain some distributional information about the depth of dependency trees. This means that despite having a higher unbiased $\mathcal{X}sp$, their true $\mathcal{X}sp$ is still smaller than that of $M_i$ before $M_8$.

From the above conclusions, we can further speculate about the workings of pretrained language models such as BERT, and we identify some related open problems.

Based on c5 and c6, we can speculate that the final layer of a pretrained language model needs to consider language information at various levels, but its memory capacity is limited. Therefore, it relies on preceding submodules to filter the information. The earlier submodules in the model encode the most generic (unbiased) structures present in the language features. As the model advances, the intermediate submodules start incorporating preferences for general structures based on actual data. Once a certain stage is reached, the later submodules in the model start to loosen their encoding of generic structures. However, due to the preference information passed from the intermediate submodules, the later submodules can still outperform the earlier submodules in encoding real structures, rather than generic ones.

Based on c3 and c6, it appears that true $\mathcal{X}sp \leq$ unbiased $\mathcal{X}sp < \mathcal{X}essp$. This suggests that for BERT, unbiased $\mathcal{X}sp$ serves as a tighter upper bound for $\mathcal{X}sp$, and there exists a submodule that achieves this upper bound. Now, the question arises: Is this also the case for general pretrained models? If so, what are the underlying reasons?

## 4.2 Using $\mathcal{X}ssp$ and $\mathcal{X}essp$ as Regularization Loss in Fine-tuning BERT

Let us denote the downstream task loss as $T(M_{24})$. Taking $\mathcal{X}ssp$ as an example, using $\mathcal{X}ssp$ as a regularizing loss during fine-tuning refers to replacing the task loss with:

$$T(M_{24}) + \lambda * \mathcal{X}ssp(M_i) \quad (1 \leq i \leq 24)$$

where $\lambda$ is a regularization parameter. The purpose of this approach is to explore the potential for enhancing the fine-tuning performance by improving the submodules of BERT in their ability to linearly encode tree structures. If there exists a submodule that achieves both enhancement in linear encoding capabilities and improved fine-tuning performance, it implies that the parameter space of this submodule, which has better linear encoding abilities, overlaps with the optimization space of fine-tuning. This intersection is smaller than the optimization space of direct fine-tuning, reducing susceptibility to local optima and leading to improved fine-tuning results.

Conversely, if enhancing certain submodules hinders fine-tuning or even leads to its failure, it suggests that the submodule's parameter space, which has better linear encoding abilities, does not overlap with the optimization space of fine-tuning. This indicates that the submodule has already attained

the smallest $\mathcal{X}ssp$ value that greatly benefits the BERT's performance.

Based on f1, we can infer that $M_{14}$ to $M_{18}$ are not suitable as enhanced submodules. According to c5, the submodules most likely to improve fine-tuning performance after enhancement should be near $M_8$. We conducted experiments on a single-sentence task called the Corpus of Linguistic Acceptability (CoLA) (Warstadt et al., 2019), which is part of The General Language Understanding Evaluation (GLUE) benchmark (Wang et al., 2019).

The test results are shown in Table 3. As predicted earlier, enhancing the submodules around $M_{14}$ to $M_{18}$ (now expanded to $M_{12}$ to $M_{19}$) proves to be detrimental to fine-tuning, resulting in failed performance. However, we did observe an improvement in fine-tuning performance for the submodule $M_{10}$ near $M_8$ after enhancement. This gives us an intuition that if we have additional topological probes and similar metrics to $\mathcal{X}ssp$ and $\mathcal{X}sp$, we can explore enhancing submodules that are in the rising phase of true $\mathcal{X}sp$, away from the boundary of unbiased $\mathcal{X}sp$ and $\mathcal{X}ssp$, in an attempt to improve fine-tuning outcomes.

| Method | mean | std | max |
|---|---|---|---|
| DF | 63.34 | 1.71 | 66.54 |
| EH $M_3$ | 63.90 | 2.66 | 68.73 |
| EH $M_5$ | 63.90 | 1.36 | 66.04 |
| EH $M_{10}$ | **64.87** | 2.07 | **68.47** |
| EH $M_{12}{\sim}M_{19}$ | 0.00 | 0.00 | 0.00 |
| EH $M_{20}$ | 5.48 | 16.46 | 54.87 |
| EH $M_{24}$ | 40.43 | 26.60 | 62.52 |

Table 3: Direct fine-tuning and sub-module enhancement test scores. Here, "DF" denotes direct fine-tuning, while "EH $M_i$" represents the fine-tuning with the enhancement of $M_i$ based on $\mathcal{X}ssp$. The evaluation metric used in CoLA is the Matthew coefficient, where a higher value indicates better performance.

## 5 Conclusion

Consider a thought experiment where there is a planet in a parallel universe called "Vzjgs" with a language called "Vmtprhs". Like "English", "Vmtprhs" comprises 26 letters as basic units, and there is a one-to-one correspondence between the letters of "Vmtprhs" and "English". Moreover, these two languages are isomorphic under letter permutation operations. In other words, sentences in "English" can be rearranged so that they are equivalent to sentences in "Vmtprhs", while preserving the same

meaning. If there were models like BERT or GPT in the "Vzjgs" planet, perhaps called "YVJIG" and "TLG," would the pretraining process of "YVJIG" on "Vmtprhs" be the same as BERT's pretraining on "English"?

In theory, there should be no means to differentiate between these two pretraining processes. For a blank model (without any training), extracting useful information from "Vmtprhs" and "English" would pose the same level of difficulty. However, it is true that "Vmtprhs" and "English" are distinct, with the letters of "Vmtprhs" possibly having different shapes or being the reverse order of the "English" alphabet. Therefore, we can say that they have different letter features, although this feature seems to be a mere coincidence. In natural language, there are many such features created by historical contingencies, such as slang or grammatical exceptions. Hence, when we aim to interpret the mechanisms of these black-box models by studying how language models represent language-specific features, we must consider which features are advantageous for interpretation and what we ultimately hope to gain from this research.

This paper presents a thorough exploration of a key issue, specifically examining the articulation of internal feature constraints. By enveloping the original feature within a feature space that adheres to such constraints, it is possible to effectively eliminate any unintended or accidental components. Within this explicitly defined feature space, metrics such as $\mathcal{X}ssp$ and $\mathcal{X}essp$ can be defined. We can subsequently examine the evolution of these metrics within the model to gain a deeper understanding of the encoding strategies employed by the model for the original feature, as described in the experimental section of this paper. Once we understand the encoding strategies employed by the model, we can investigate the reasons behind their formation and the benefits they bring to the model. By conducting studies on multiple similar features, we can gain a comprehensive understanding of the inner workings of the black box.

## Limitations

The main limitation of this research lies in the approximate construction of $mins_W$ and $maxs_W$, which leads to true $-log(\mathcal{X}sp)$ surpassing $-log(\mathcal{X}ssp)$ near $M_{16}$ to some extent. However, this may also be due to their proximity, resulting in fluctuations within the training error. On

the other hand, the proposed construction scheme for the topological probe discussed in this paper lacks sufficient mathematical formalization. One possible approach is to restate it using the language of category theory.

## Acknowledgements

We thank the anonymous reviewers for their helpful comments and suggestions. This work was supported by National Natural Science Foundation of China (Nos. 62362015, 62062027 and U22A2099) and the project of Guangxi Key Laboratory of Trusted Software.

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

# A  Proof of Theorem 1

*Proof.* For any sequence $x \in X_{C_{dep}}$ that is in the same order as $pdep_H$, according to the inequality of rankings, for any permutation $\pi_x$ of $x$, we have:

$$\sum_{i=1}^{L_H} \pi_x(w_i) * \boldsymbol{p}dep(h_i) \leq \sum_{i=1}^{L_H} x(w_i) * \boldsymbol{p}dep(h_i)$$

Therefore,

$$\sum_{i=1}^{L_H} (\pi_x(w_i) - \boldsymbol{p}dep(h_i))^2$$

$$\geq \sum_{i=1}^{L_H} (x(w_i) - \boldsymbol{p}dep(h_i))^2$$

Since $pesu_W$ and $\boldsymbol{p}dep_H$ are in the same order, we just need to prove that any sequence $x \in X_{C_{dep}}$ and in the same order as $\boldsymbol{p}dep_H$ satisfies

$$\sum_{i=1}^{L_H} (x(w_i) - \boldsymbol{p}dep(h_i))^2$$

$$\geq \sum_{i=1}^{L_H} (pesu(w_i) - \boldsymbol{p}dep(h_i))^2$$

The theorem is automatically established. Because

$$\sum_{i=1}^{L_H} (pesu(w_i) - \boldsymbol{p}dep(h_i))^2$$

$$= \sum_{i=1}^{L_H} (pre(w_i) - \boldsymbol{a}\boldsymbol{p}dep_i)^2 \qquad (18)$$

, without loss of generality, we can assume $pesu_W$ and $x$ to be ascending sequences and not equal and exist a $k$ such that when $1 \leq i \leq k-1$

$$pesu(w_i) = x(w_i) \qquad (19)$$

and

$$pesu(w_k) \neq x(w_k) \qquad (20)$$

Based on the recursive condition, we can infer that

$$|pesu(w_k) - x(w_k)| = 1 \qquad (21)$$

Combined with the value condition of $bias_{k-1}$, we further find that

$$|pesu(w_k) - \boldsymbol{ap}dep_k| \le |x(w_k) - \boldsymbol{ap}dep_k|$$

The inductive hypothesis when $i = m$ is

$$|pesu(w_m) - \boldsymbol{ap}dep_m| \le |x(w_m) - \boldsymbol{ap}dep_m|$$

Due to the condition $\boldsymbol{ap}dep_{m+1} - \boldsymbol{ap}dep_m \le 1$ and the value condition of $bias_m$, it still holds when $i = m + 1$ that

$$|pesu(w_{m+1}) - \boldsymbol{ap}dep_{m+1}|$$
$$\le |x(w_{m+1}) - \boldsymbol{ap}dep_{m+1}|$$

Thus, when $i \ge k$

$$(x(w_i) - \boldsymbol{p}dep(h_i))^2 \ge (pesu(w_i) - \boldsymbol{p}dep(h_i))^2.$$

$\square$

## B   Analysis of Tree Depth Minimum

The minimum of $pesu_W$ is denoted as $dep_{min}$. Fixing $pesu_W$, we let all sets (or sequences) of vectors satisfying the following conditions compose a set denoted by $\Omega_{pesu_W}$.

$$\exists\, P \in R^{m*n},\ \forall\, i(i = 1, 2, \cdots L_H)$$

$$pesu(w_i) - \sqrt{\epsilon_i} < h_i^T P^T P h_i < pesu(w_i) + \sqrt{\epsilon_i}$$

Here, $\epsilon_i \ll pesu(w_i)^2$. Let $pesu(w_1)$ be $dep_{min}$ and $pesu(w_i) \le pesu(w_{i+1})(i = 1, 2, \cdots L_H - 1)$ without loss of generality, and the following theorem can be obtained.

**Theorem 2.** *For any two different sequences $pesu_W$ and $pesu'_W$, if $pesu(w_1) > 0$ and $pesu'(w_1) > 0$. there is a one-to-one mapping $\phi$ between $\Omega_{pesu_W}$ and $\Omega_{pesu'_W}$.*

*Proof.* We construct $\phi$ such that

$$\forall H \in \Omega_{pesu_W}$$

$$\phi(H) = (h_1', h_2', \cdots h_{L_H}') = H' \in \Omega_{pesu'_W}$$

Here, $h_1' = h_1$, and when $i = 2, 3 \cdots, L_H$

$$h_i' = \frac{\sqrt{pesu'(w_i) * pesu(w_1)}}{\sqrt{pesu(w_i) * pesu'(w_1)}} h_i$$

Since

$$\exists P \in R^{m*n}, \forall i(i = 1, 2, \cdots L_H)$$

$$pesu(w_i) - \sqrt{\epsilon_i} < h_i^T P^T P h_i < pesu(w_i) + \sqrt{\epsilon_i}$$

$$\epsilon_i \ll pesu(w_i)^2$$

Let $P' = \dfrac{\sqrt{pesu'(w_1)}}{\sqrt{pesu(w_1)}} P$ and when $i = 1, 2, \cdots L_H$

$$\epsilon_i' = (\frac{pesu'(w_i)}{pesu(w_i)})^2 \epsilon_i,$$

then

$$\epsilon_i' \ll (\frac{pesu'(w_i)}{pesu(w_i)})^2 pesu(w_i)^2 = pesu'(w_i)^2$$

After calculation,

$$\forall i(i = 1, 2, \cdots L_H)$$

$$pesu'(w_i) - \sqrt{\epsilon_i'}$$
$$< (h_i')^T (P')^T P' h_i'$$
$$< pesu'(w_i) + \sqrt{\epsilon_i'}$$

Therefore, $\phi$ is well defined, and $\forall H_i, H_j \in \Omega_{pesu_W}$ when $H_i \ne H_j$

$$\phi(H_i) \ne \phi(H_j)$$

Therefore, $\phi$ is also an injective function. It is easy to prove that the inverse map $\phi^{-1}$ of $\phi$ is also an injective function and satisfies the above conditions. $\square$

The proof of the theorem above does not apply to the cases where $pesu(w_1) = 0$ or $pesu'(w_1) = 0$. If $dep_{min}$ is greater than 0, then the results of the tree topological probe do not necessarily depend on the selection of $dep_{min}$, and we may set it as 1. However, we have not further explored whether Theorem 2 is necessarily invalid. Nevertheless, we can examine the drawbacks that arise from setting $dep_{min}$ to 0 from another perspective.

When $i \ge 2$, $h_i$ is projected by $P$ near the $(m)$-dimensional sphere with a radius of $\sqrt{pesu(w_i)}$,

$$\forall i = 1, 2 \cdots, L_H$$

$$|h_i^T P^T P h_i - pesu(w_i)| < \epsilon_i$$

If $dep_{min} = 0$, then the topology of the geometric space composed of all vectors $Ph_1$ satisfying $|h_1^T P^T P h_1 - dep_{min}| < \epsilon_1$ is homeomorphic to an $m$-dimensional open ball. This may result in probes exhibiting different preferences for the root and other nodes. However, if $dep_{min} > 0$, the topology of the geometric space is an $m$-dimensional annulus, which is the same for all nodes, thus avoiding the issue of preference.

## C   Data for Training and Evaluating Probes

To ensure the reliability and diversity of data (appropriate sentences) sources, we separated the sentences participating in the probe experiment from the training, verification and test data sets of some tasks of The General Language Understanding Evaluation (GLUE) benchmark (Wang et al., 2019).

We selected four small sample text classification tasks in GLUE with reference to (Hua et al., 2021), namely, the Corpus of Linguistic Acceptability (CoLA) (Warstadt et al., 2019), Microsoft Research Paraphrase Corpus (MRPC) (Dolan and Brockett, 2005), Recognizing Textual Entailment (RTE) (Wang et al., 2019) and Semantic Textual Similarity Benchmark (STS-B) (Cer et al., 2017), which cover the three major task types of SINGLE-SENTENCE, SIMILARITY AND PARAPHRASE and INFERENCE in GLUE. MRPC, RTE and STS-B are all double sentence tasks, and the experiment needs only BERT to represent a single sentence; thus, we consider two sentences that belong to the same group of data independently, not spliced.

After the data sets of the four tasks are processed as above, the remaining statements are merged into a raw text data set $rtd_{mix}$, which contains 47136 sentences. This is close to the size of the Pennsylvania tree database (Marcus et al., 1993) used by the structural probe (Hewitt and Manning, 2019); short and long sentences are evenly distributed.

## D   Experimental Setup for Training Probes and Fine-tuning

We use the BERT implementation of Wolf et al. and set the rank of the probe matrix to be half the embedding dimension. The probe matrix is randomly initialized following a uniform distribution $U(-0.05, 0.05)$.

We employ the AdamW optimizer with the warmup technique, where the initial learning rate is set to 2e-5 and the epsilon value is set to 1e-8. The training stops after 10 epochs. The training setup for fine-tuning experiments is similar to that of training probes. One notable difference is the regularization coefficient $\lambda$, which is dynamically determined after one epoch of training, ensuring that $\frac{\lambda * \mathcal{X}_{ssp}(M_i)}{T(M_{24})} \approx 0.1$, without any manual tuning.

We conduct experiments on each fine-tuning method by using 10 different random seeds, and we compute the mean, the standard deviation (std), and the maximum values.

## E   Supplementary Chart Materials

Table 4 lists the exact measurements of $\mathcal{X}ssp$, $\mathcal{X}essp$, and true $\mathcal{X}_{sp}$ for BERT-Large.

| $M$ | $\mathcal{X}_{ssp}$ | $\mathcal{X}_{essp}$ | $\mathcal{X}_{tsp}$ |
|---|---|---|---|
| $M_0$ | 0.039 | 5.382 | 0.3084 |
| $M_1$ | 0.017 | 0.536 | 0.2644 |
| $M_2$ | 0.017 | 0.526 | 0.244 |
| $M_3$ | 0.018 | 0.348 | 0.2016 |
| $M_4$ | 0.033 | 0.351 | 0.1701 |
| $M_5$ | 0.025 | 0.52 | 0.1622 |
| $M_6$ | 0.023 | 0.52 | 0.1559 |
| $M_7$ | 0.013 | 0.345 | 0.14 |
| $M_8$ | 0.01 | 0.347 | 0.1424 |
| $M_9$ | 0.011 | 0.352 | 0.1577 |
| $M_{10}$ | 0.013 | 0.359 | 0.1415 |
| $M_{11}$ | 0.021 | 0.375 | 0.1128 |
| $M_{12}$ | 0.054 | 0.391 | 0.0975 |
| $M_{13}$ | 0.076 | 0.42 | 0.0764 |
| $M_{14}$ | 0.084 | 0.467 | 0.0651 |
| $M_{15}$ | 0.088 | 0.525 | 0.0616 |
| $M_{16}$ | 0.09 | 0.663 | 0.0656 |
| $M_{17}$ | 0.086 | 0.785 | 0.0808 |
| $M_{18}$ | 0.09 | 0.883 | 0.1155 |
| $M_{19}$ | 0.09 | 0.999 | 0.1416 |
| $M_{20}$ | 0.092 | 1.045 | 0.1615 |
| $M_{21}$ | 0.094 | 1.447 | 0.2468 |
| $M_{22}$ | 0.102 | 1.715 | 0.28634 |
| $M_{23}$ | 0.107 | 1.709 | 0.3171 |
| $M_{24}$ | 0.113 | 1.837 | 0.328 |

Table 4:  Exact values of $\mathcal{X}_{ssp}$, $\mathcal{X}_{essp}$, and true $\mathcal{X}_{sp}$ for $M_i$

## F   Experimental data for RoBERTa-large

Figure 3 shows the negative logarithm curves of three measurement values as a function of variation in Mi for RoBERTa-Large. Table 5 lists the exact measurements of $\mathcal{X}ssp$, $\mathcal{X}essp$, and true $\mathcal{X}_{sp}$ for RoBERTa-Large.

From the experimental data, it is evident that the overall pattern of evolution in the graphs for RoBERTa-Large and BERT-Large is consistent. There's a slight initial increase followed by a decline, but the boundaries for $\mathcal{X}sp$ in the case of RoBERTa-Large are much tighter, especially in the earlier modules.

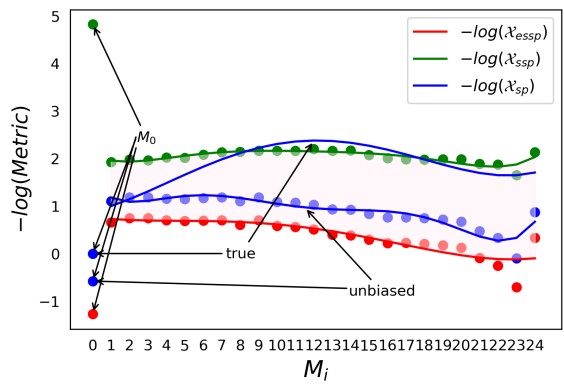

Figure 3: Negative logarithm of $\mathcal{X}_{ssp}$, $\mathcal{X}_{essp}$, unbiased $\mathcal{X}_{sp}$ and true $\mathcal{X}_{sp}$ across $M_i$.

| $M$ | $\mathcal{X}_{ssp}$ | $\mathcal{X}_{essp}$ | $\mathcal{X}_{tsp}$ |
|---|---|---|---|
| $M_0$ | 0.008 | 3.532 | 0.991 |
| $M_1$ | 0.145 | 0.515 | 0.243 |
| $M_2$ | 0.137 | 0.469 | 0.446 |
| $M_3$ | 0.139 | 0.470 | 0.331 |
| $M_4$ | 0.131 | 0.493 | 0.257 |
| $M_5$ | 0.132 | 0.500 | 0.199 |
| $M_6$ | 0.123 | 0.494 | 0.153 |
| $M_7$ | 0.117 | 0.491 | 0.110 |
| $M_8$ | 0.117 | 0.542 | 0.109 |
| $M_9$ | 0.114 | 0.491 | 0.087 |
| $M_{10}$ | 0.113 | 0.555 | 0.091 |
| $M_{11}$ | 0.113 | 0.567 | 0.091 |
| $M_{12}$ | 0.110 | 0.598 | 0.094 |
| $M_{13}$ | 0.114 | 0.667 | 0.101 |
| $M_{14}$ | 0.113 | 0.675 | 0.102 |
| $M_{15}$ | 0.124 | 0.738 | 0.120 |
| $M_{16}$ | 0.133 | 0.797 | 0.136 |
| $M_{17}$ | 0.136 | 0.789 | 0.131 |
| $M_{18}$ | 0.137 | 0.807 | 0.136 |
| $M_{19}$ | 0.136 | 0.831 | 0.135 |
| $M_{20}$ | 0.136 | 0.880 | 0.145 |
| $M_{21}$ | 0.150 | 1.086 | 0.169 |
| $M_{22}$ | 0.153 | 1.277 | 0.176 |
| $M_{23}$ | 0.190 | 2.006 | 0.197 |
| $M_{24}$ | 0.117 | 0.711 | 0.201 |

Table 5: Exact values of $\mathcal{X}_{ssp}$, $\mathcal{X}_{essp}$, and true $\mathcal{X}_{sp}$ for $M_i$