# OpenReview forum: "Rethinking the Construction of Effective Metrics for Understanding the Mechanisms of Pretrained Language Models"
_EMNLP/2023/Conference — EMNLP 2023 Findings_

### Official Review · Reviewer_cpmL · 2023-08-04

**Soundness:** 3

**Excitement:**

3: Ambivalent: It has merits (e.g., it reports state-of-the-art results, the idea is nice), but there are key weaknesses (e.g., it describes incremental work), and it can significantly benefit from another round of revision. However, I won't object to accepting it if my co-reviewers champion it.

**Missing References:**

Unsupervised Probing (also with syntactic structure)
- https://aclanthology.org/2020.acl-main.383.pdf

Causal Probing
- https://direct.mit.edu/tacl/article/doi/10.1162/tacl_a_00359/98091/Amnesic-Probing-Behavioral-Explanation-with
- https://direct.mit.edu/coli/article/47/2/333/98518


**Paper Topic And Main Contributions:**

The paper is about the interpretability of NLP models, specifically in the domain of probing classifiers for linguistic properties.
The authors propose a new approach for designing an unsupervised probe, reminiscent of the probe proposed by Hewitt and Manning for recovering syntactic structure, but unsupervised.
The authors define such a probe (again, similar to the design of Hewitt and Manning, by trying to replicate a syntactic tree structure, defined by the depths of the nodes (words/tokens) in the graph) quite thoroughly, discussing the different properties and constraints on such tree, and how to achieve (approximate) it.

*Main Contributions*

- The authors propose a new unsupervised structural probe for analyzing the internal representations of neural language models
- Using a metric defined for such a probe, the authors propose a method to improve fine-tuning.

*Methodology*

While my topology background is not very strong, I did my best to follow the different metrics, steps, and logical steps made in the paper, and it sounds reasonable (although I cannot guarantee that).

*Evaluation*

The paper conducts two main experiments:
1. Measuring their probe using BERT-large on different sentences (from different GLUE datasets)
2. Using their derived metric as an additional loss function during fine-tuning of the model on the linguistic acceptability dataset (CoLA).

The first experiment was unclear to me. I could not understand the results or the claims that the authors made.
The second experiment shows a minor improvement in performance and is not convincing. Moreover, the authors did not perform any statistical significance tests, which makes it hard to take the 1.5 difference very seriously.
Besides these two experiments, there were no others.
I don’t understand this lack of comparison, given the wide range of relevant papers in the field (which they also cite).
How does their probe compare to others? E.g. Hewitt and Manning
How does their probe compare to the “real” syntactic tree of a sentence (using an annotated treebank)

*Writing*

The paper was very hard to understand.
It makes very strong claims, which I often couldn’t understand or interpret. It is very opaque at times. The results and their interpretation as a list of findings (written as fN.) and conclusions (written as cN.), open problems (written as sN.), and preferences (written as pN.) are also nonstandard. Such writing is not problematic per se, but it breaks the flow of the paper and reads as a list of findings, rather than a story, which is much harder to understand. Specific details are given in the “presentation improvements” section.
Overall recommendation: while I am generally excited about the idea and direction of the paper, I think the current state of writing (and experiments, to some degree) can be greatly improved. I strongly encourage the authors to resubmit the paper in an improved state.


**Reasons To Accept:**

- The paper provides an interesting formal explanation of probing, and specifically the purpose of unsupervised probing
- The paper proposes a new unsupervised structural probe


**Reasons To Reject:**

- The paper’s experimental setup is lacking comparison with previous work, and evaluation of the constructed probe
- The paper’s writing can be greatly improved


**Reproducibility:**

3: Could reproduce the results with some difficulty. The settings of parameters are underspecified or subjectively determined; the training/evaluation data are not widely available.

**Reviewer Confidence:**

3: Pretty sure, but there's a chance I missed something. Although I have a good feel for this area in general, I did not carefully check the paper's details, e.g., the math, experimental design, or novelty.

**Typos Grammar Style And Presentation Improvements:**

- L.1 LMs are *not* expected to effectively map input text to vectors, this is perhaps an implicit requirement, but their main objective is to be able to predict the next word in a sequence
- L.3 what are inherent relationships?
- L.6 what are internal relations?
- L.10 what do you mean by white-box?
- L.12 what do you mean by inherent interpretability?
- L.15 what do you mean by a novel line?
- L.19 “line of investigation” sounds peculiar in this context
- L.21 “conducted measurements” sounds peculiar
- L.23 “propose a speculation”
- L.24 what is that “working mechanism”?
- L.28 what are submodules?
- L.34 “algebraic operation” is very vague
- L.37 what do you mean by “what-box model” in this context?
- L.45 “metric computation process usually requires interpretability” - strange phrasing, I cannot make sense of it
- L.47 “inherent interpretability of the source model is missing” - strange phrasing, I cannot make sense of it
- L.50 what do you mean by semiquantitative?
- L.51 what are artificial hypotheses?
- L.54 what’s the connection here between contextual embeddings and static embeddings?
- L.56 what does intuition have to do with interpretability?
- L.62 what do you mean by “complete word”?
- L.71 what are “complete syntax relations”?
- L.76 “exist other embeddings” - I don’t see the connection
- L.79 why do we care about an upper bound?
- L.117 how is your approach “beneficial for interpretability”?
- L.148 what are submodules?
- L.181 what is “Model Evidence Maximum”?
- The presentation of the equations can also be improved to be more friendly (e.g. pdep -> p-dep)
- Tables 1+2: they are not clear, and their captions are also very opaque.
- L.425-434 (e1-e4) unclear experiments
- L.444 strange formatting (f1.)
- L.542-546 I don’t understand this claim and why would we want these properties to be linearly encoded in the representations
- L.607-625. This paragraph is very opaque

---

> ### Author Rebuttal · Authors · 2023-08-28
>
> Dear Reviewer cpmL,
>
> Thank you once again for your comments and the questions you've raised. We have included the supplementary experiments in the official comment.
>
> Regarding the issue you mentioned about comparing probes, it's possible that there might be a misunderstanding about the problem being discussed in the paper and the proposed methodology. This paper doesn't aim to introduce a probe that possesses the same functionality as the structural probe but is superior, but it presents the concept of reasonably constructing upper and lower bounds for metrics. The ssp and essp introduced in the paper are simply instances where we apply this concept to the structural probe, demonstrating that this idea is not only theoretical but also actionable.
>
> There isn't a direct correspondence between a probe and its upper and lower bound probes in terms of labels. There might be some relationship between them, but it's not yet clear. Therefore, comparing mins_{W} with the actual depth of the dependency tree currently lacks significance. The primary purpose of providing upper and lower bounds for metrics is to further enhance the interpretability of the metrics and to construct theoretical analytical models on metrics that require the use of additional labels for computation. This aspect can be observed from the section 3.4 and Experiment 1.  Moreover, metrics that require the use of additional labels for computation cannot serve as universal regularized loss functions.
>
> Regarding the questions you raised in the "Typos Grammar Style And Presentation Improvements" section, we have provided some responses in the official comment.

---

### Official Review · Reviewer_RAmS · 2023-08-04

**Soundness:** 3

**Excitement:**

3: Ambivalent: It has merits (e.g., it reports state-of-the-art results, the idea is nice), but there are key weaknesses (e.g., it describes incremental work), and it can significantly benefit from another round of revision. However, I won't object to accepting it if my co-reviewers champion it.

**Paper Topic And Main Contributions:**

The paper is focused on designing a suite of metrics called the tree topological probe which is used for interpretability of pre-trained language models. Specifically, the paper examines BERT-large and uses their findings to enhance fine-tuning performance. The main contribution of the work is to provide the interpretability community with a self-supervised probe to examine (pre-trained) language models.

**Reasons To Accept:**

I like the general intuitions you have on the probe as detailed in section 3.1. The work has theoretical contributions and the framework overall could be beneficial to the wider NLP interpretability community.

The findings in sections 4.1 are interesting and reveal insights previously not available in the structural probe alone. However, it's really difficult to interpret how these findings could apply to the broader community as it is only performed on BERT-large. The subsequent results of fine-tuning layers using insights from 4.1 are convincing.

**Reasons To Reject:**

My main worries for the paper are:
1. The limited empirical evidence of the work given that the main experiments were only conducted on a single model. I think having two or three additional analyses of related models could really strengthen your work.

2. There are a number of places where perturbations/sensitivity analysis could be done to reveal the robustness of the method/results. Calculating error bounds in Table 3, using multiple different datasets, different ways of approximating max(s_w).

3. The writing could benefit from including more motivation throughout the introduction and transition paragraphs. Why is it important study these phenomena? How would this benefit the larger community? I think there are compelling reasons here and the authors should seek to articulate them.

4. The ethics statement feels detached/removed from the rest of the paper. It's important for the ethics section to engage with the ethical harms and considerations of the presented theories and conducted experiments.

Details:
There are a number of places where the authors are making assumptions without citations or sufficient motivation and reasoning.
e.g., lines 052-059 --- it's not immediately clear to readers why the distinction between contextual and static embeddings is sufficient for researchers to trust a model's effectiveness. "Researchers' intuitions" is not defined/a poor motivation.

There is motivation provided in the introduction that is not answered throughout the paper e.g., lines 075-080. These are not central questions to the paper and would recommend cutting for space and overall tightening the writing of the paper.

There are also some claims that could be reworded with more hedges/citations as they are not obviously true statements e.g., lines 100-102. There is also a volume of follow-up work to Hewitt and Ethayarajh which should be cited here as well.

They key intuition in the self-supervised structural probe is a little hidden in the introduction: line 125 "In the case where the internal constraints of the probed features are well defined, a probe that detects these features can naturally induce a probe that detects the internal constraints, which is self-supervised." This should be highlighted earlier in the introduction.

A key challenge which you hint on line 227 is that there are multiple ways of constructing model M. It would be nice to see more evaluation here (ablation studies) to understand how sensitive the probe is to these chosen parameters.

Writing suggestions:
It would be nice if you could define min(s_w) and max(s_w) at the same place in section 3.3 rather than defining max(s_w) when you start talking about enhancements. It would also be nice to intuitively describe what each of these sequences represent. From there, you can easily present equations (0), (10), and (12).

Having language like, "since" and "similarly" is not sufficient language to transition between your equations line 366. Why is it important to present the following equations? Why do we care about the properties of these sets? The motivation here is lost to the reader and could really strengthen your work.

In the case where Xssp(M) = Xessp(M), it would be interesting to know how often this happens. You could put a nice metric to how often the structural probe would miss insights only your probe can interpret. Could also consider moving content from lines 389-403 together with this initial definition (could also go in the appendix).

Line 414, need some justification on why you are focused on BERT-large. Is it the most representative model? Currently as it reads, it's difficult to know if this only works on BERT-large or if there are unique properties to BERT-large that make this analysis rich. Do you have insights on what woudl happen with BERT-base/Roberta?

**Reproducibility:**

3: Could reproduce the results with some difficulty. The settings of parameters are underspecified or subjectively determined; the training/evaluation data are not widely available.

**Reviewer Confidence:**

3: Pretty sure, but there's a chance I missed something. Although I have a good feel for this area in general, I did not carefully check the paper's details, e.g., the math, experimental design, or novelty.

**Typos Grammar Style And Presentation Improvements:**

This is a nit but equation (1) is really critical to the understanding of the intuition of your paper. It would be helpful to try and be very clear with the language leading up to it, sections 3.1 and 3.2. A simple figure could also help.

Lines 307 - 323 are dense to read. It might help to define and set up all the variables first, before jumping into equations. They currently read as sentences which is hard to parse.

Line 382 could be moved to the appendix. Might be nice to have 367 as a separate section that is comparing and contrasting to the structural probe.

---

> ### Author Rebuttal · Authors · 2023-08-28
>
> Dear reviewer QnWV
>
> Thank you for providing many specific suggestions on improving paper writing. We will consider how to make the paper more reader-friendly in terms of readability.
>
> In response to the technical issues raised in your review comments, we have addressed and included supplementary experiments in the official comment. Lastly, concerning the writing issues, we would like to make some subtle points of defense.
>
> Firstly, it's undeniable that this paper does not present any ethical concerns in terms of theory and methodology. The discussion in the "Ethics Statement" section is based on the notion of "potential societal impact." While it's possible that the actual content of this discussion may not perfectly align with the Ethics Statement section, it is not disjointed from the rest of the paper's content. Currently, there are indeed numerous voices expressing concerns about the LLMs. Therefore, it's reasonable to "inspire further reflections on the advantages of human intelligence compared to the intelligence of these models," as stated in the paper. According to the language used in this paper, if the models excel over humans in the sp aspect, could they potentially outperform humans in the essp or ssp aspects as well? Furthermore, what are the distinguishing factors between possessing essp or ssp capabilities and having sp capabilities?

---

### Official Review · Reviewer_QnWV · 2023-08-05

**Typos Grammar Style And Presentation Improvements:** L274, given should be in \text{given}…
**Soundness:** 4

**Excitement:**

5: Transformative: This paper is likely to change its subfield or computational linguistics broadly. It should be considered for a best paper award. This paper changes the current understanding of some phenomenon, shows a widely held practice to be erroneous in someway, enables a promising direction of research for a (broad or narrow) topic, or creates an exciting new technique.

**Paper Topic And Main Contributions:**

This paper rethinks the probing of PLM representations.
Based on the motivation that the current probing methods lack an upper bound to help us understand the model capability more clearly, the authors propose a novel self-supervised probe: a tree-topological probe to probe the hierarchical structure learned by BERT. They theoretically justify the bounding relationship of the tree-topological probe and the structural probe and apply the probing method on BERT-large. Based on their probing results, they provide several findings and speculation, and the probing method can help find the submodules that need to be enhanced. They verify these findings by finetuning experiments, their probing method indeed can identify the submodules that need to (or cannot) be enhanced (by structural knowledge).

**Questions For The Authors:**

- Is your method potentially applicable to LLMs?
- Will the distribution approximation at L383 significantly affect the probing results?

**Reasons To Accept:**

- The motivation is great, existing probing methods usually can only provide results like "A is better than B", and the results are often predictable. This paper proposes a probing method that can know "Is A or B good enough?", which can provide additional insights into the model's capability.
- The paper provides a detailed theoretical analysis and a "Rethinking process" on how to induce the self-supervised probe.
- They proposed the tree-topological probe that is "upgraded" from the structural probe, and the new probing method can provide an upper bound and has the potential to be a guideline for "upgrading" other probing methods.
- Authors provide extensive analysis of the probing results and produce findings and speculations.
- Authors verify their findings by a finetuning experiment, showing that their probing method is useful to identify the capability of each submodule, and enhancing these submodules by hierarchical information is useful. Which makes the whole paper self-consistent.

**Reasons To Reject:**

- Would be better by more experiments (as you mentioned, e2,3,4, and other PLMs)


**Reproducibility:**

5: Could easily reproduce the results.

**Reviewer Confidence:**

3: Pretty sure, but there's a chance I missed something. Although I have a good feel for this area in general, I did not carefully check the paper's details, e.g., the math, experimental design, or novelty.

---

> ### Author Rebuttal · Authors · 2023-08-28
>
> Dear reviewer QnWV，
>
> Thanks again for your affirmation of our research.
>
> We have included additional experiments and responses to questions 1 and 2 in the official comments. The experimental results align with the findings and conclusions stated in our paper, and by examining the results of the supplementary experiments, we can observe other interesting phenomena. Unfortunately, experiments on Albert-large have not been completed at this time.
>
> However, despite the further experimental validation, our intention was not to make strong assertions that the findings and conclusions in the paper hold true for all PLMs. These are ultimately intuitions and not rigorously substantiated. The core of this paper lies in the analysis from sections 3.1 to 3.4 and the ideas we aim to convey in the conclusion. These aspects are rigorous.

---

### Meta-Review · Area_Chair_c2Ce · 2023-09-19

**Recommendation:** 3

**Metareview:**

This paper proposes a new method (a tree topological probe) to probe syntactic/hierarchical structure learned by BERT in an unsupervised manner. The authors use their findings to also enhance fine-tuning performance of BERT.

Pros:
- Reviewers found the paper has strong potential and addresses an important problem in the probing area.
- They found the proposed method interesting and appreciated the theoretical contributions.
- They did not have any critique of the method itself, though this may be due to a lack of understanding (see below).

Cons:
- Multiple reviewers felt that only experimenting on BERT-Large was a limitation. During the author response, the authors provided more results on RoBERTa-Large and Albert-Large to alleviate this concern.
- [**important**] Multiple reviewers highlighted that the paper was lacking clarity & very difficult to read. They felt the paper was missing a core narrative structure to motivate the work, and that the necessary changes would take more than a minor revision to fix.

Reviewers had some disagreement over scores. Taking all of the above into account, I have adjudicated in favor of the 2 reviewers in agreement (but with overall lower scores). I think the paper has some very interesting ideas and seems generally sound, but certain parts are confusing & may need more clarification.

---

### Decision · Program_Chairs · 2023-10-07

**Decision:**

Accept-Findings

**Comment:**

This paper proposes a new method (a tree topological probe) to probe syntactic/hierarchical structure learned by BERT in an unsupervised manner. The authors use their findings to also enhance fine-tuning performance of BERT.

Pros:
- Reviewers found the paper has strong potential and addresses an important problem in the probing area.
- They found the proposed method interesting and appreciated the theoretical contributions.
- They did not have any critique of the method itself, though this may be due to a lack of understanding (see below).

Cons:
- Multiple reviewers felt that only experimenting on BERT-Large was a limitation. During the author response, the authors provided more results on RoBERTa-Large and Albert-Large to alleviate this concern.
- [**important**] Multiple reviewers highlighted that the paper was lacking clarity & very difficult to read. They felt the paper was missing a core narrative structure to motivate the work, and that the necessary changes would take more than a minor revision to fix.

Reviewers had some disagreement over scores. Taking all of the above into account, I have adjudicated in favor of the 2 reviewers in agreement (but with overall lower scores). I think the paper has some very interesting ideas and seems generally sound, but certain parts are confusing & may need more clarification.